# Investigating the Impact of Financial Inclusion Drivers, Financial Literacy and Financial Initiatives in Fostering Sustainable Growth in North India

Amit Pandey *, Ravi Kiran and Rakesh Kumar Sharma

School of Humanities and Social Sciences, Thapar Institute of Engineering and Technology, Patiala 147004, India
* Correspondence: amitpandey@thapar.edu

**Abstract:** The present study examines how successful we are in achieving financial inclusiveness, investigating the influence of the drivers of financial inclusion (FI), financial literacy, and financial initiatives on sustainable growth. The drivers of FI considered are digitalization, technology, and usage. This study proceeds with a difference and investigates the impact of the drivers on sustainable growth through the mediation of financial literacy. The basic purpose is to understand whether mediation assists in enhancing the impact of the drivers of FI on sustainable growth. Sustainable growth is measured by knowing customers' perceptions regarding FI success through the achievement of the SDGs, viz., SDGs 1, 3, 5, 8, 9, 10, 11, and 17, especially related to poverty alleviation; removing gender inequality; and promoting industrial growth. The study uses PLS-SEM modeling to investigate the impact of the drivers of FI, financial literacy, and financial initiatives on sustainable growth. The results highlight that usage, digitalization, and FinTech emerged as significant drivers of FI. The study assesses the direct impact of the drivers of FI on sustainable growth and the indirect effect through the mediation of financial literacy. This is indicative of the importance of financial literacy in accentuating the impact of the drivers on sustainable growth. However, financial initiatives positively impact sustainable growth in the northern region of India as well.

**Keywords:** financial inclusion; financial literacy; sustainable development goals (SDGs); financial policy; digitalization; technology; PLS-SEM

## 1. Introduction

Across the globe, there is an increased emphasis on FI, especially in emerging economies, with the motive to enhance economic growth and decrease poverty [1]. However, there are widespread disparities existing worldwide with regard to access to financial services [2]. Many researchers have also highlighted how financial exclusion could hinder people from leading a normal life. According to Carbo Gardener and Molyneux [3], financial access has a robust causal association with social exclusion. Claessens [4] backed this viewpoint on social exclusion. Further, Basu and Srivastava [5] found that 70% of rural marginal/farmers lacked access to bank accounts and 87% lacked access to loans. This is prevailing despite researchers' consensus that financial inclusiveness is a basic pillar of sustainable growth. To tackle the disparity of the reach of financial services to weaker sections and unbanked areas and sectors, many countries are focusing on microfinance agencies [6]. Owing to the deficient infrastructure and poor economic conditions, the rural poor in developing economies end up having lower access to financial services [7]. Bhanot et al. [8] highlighted region-wise disparity and pointed to the low level of FI in the northeast region of India. They pointed at the vital role that could be played by self-help groups (SHGs) and education to improve inclusion. As suggested by Gwalani and Parkhi [9], due to diversity and prevalent diversification, there is a need in India for a more innovative and developed model for growth. Sharma [10] indicated bank branch penetration, availability, and the affordability of financial/banking services as the main dimensions of FI. Liu and

Walheer [11] stress the importance of catching-up effects for countries with lower levels of FI. The authors also claim that governments have improved the climate for FI in the majority of countries. Despite this, the magnitude is relatively smaller; hence, more efforts are needed. Hence, a sustainable development goal (SDGs) has been introduced to achieve financial inclusiveness and sustainable growth in society.

According to a number of studies, poverty and a lack of knowledge about financial services have been shown as the major barriers to formal financial services access. Financial literacy is the possession of knowledge of fundamental financial concepts to manage financial resources [12,13]. Financial literacy assists in the acquisition of skills essential for financial efficiency. However, it is financial knowledge, along with financial competencies, which will help to provide not only the "ability to act" but also an "opportunity to act", Huang et al. [14]. There is a need to examine how financial literacy can be related to achieving FI and sustainable growth. Many financial initiatives and policy change programs were undertaken in India to enhance FI and the economy's growth. In 2014, the government of India commenced Pradhan Mantri Jan Dhan Yojana (PMJDY) for attaining effective FI. As indicated by Poonam and Chaudhry [15], the attainment of FI has improved in many states. Despite this, the country's large populace is still excluded from the formal financial system [16]. Thus, in view of this, it is important to gauge the perception of bank customers to analyze how they relate the success of these initiatives and policies and associate it with sustainable growth.

Thus, our research is figuring out how FI is linked with sustainable growth, which is a crucial question demanding the attention of researchers. A few researchers investigating the relationship have suggested a strong association between financial development and economic growth [17]. Researchers such as Klapper et al. [18] indicate that FI enhances accessibility to credit, encourages investment facilitation along with the entry of new firms and thus improves economic growth. In the long run, FI could generate employment opportunities and ensure economic and financial stability [19]. Wang and Guan [20] highlighted the need for a sound financial system and considered financial literacy and communication technology as important determinants of FI. Greater FI may help to promote inclusive and sustainable economic development, which may result in poverty alleviation along with economic and social growth of the economy [21].

The current FI argument is based on the belief that inclusive financial institutions help people escape poverty by stimulating economic development in their societies [22]. Therefore, to overcome the issue of poverty, the Indian government, with the support of the reserve bank of India, prepared the National Financial Inclusion Strategy (NFIS). The Pradhan Mantri Jan Dhan Yojna (PMJDY) plan was also propelled in 2014 to empower the under-banked/unbanked people [23]. The United Nations sustainable development goals (SDGs) indicate FI as a crucial facilitator for sustainable growth. The United Nations SDGs policy has 17 significant objectives. SDGs 1, 2, 5, 8, and 9 are directly related to FI. SDG-1 stresses that the more inclusive a country's financial institutions are, the more capable its poorer portions will be in achieving their economic aspirations, such as establishing new enterprises and increasing their children's non-cognitive and cognitive development [24]. SDG-2 indicates that financially included farmers can make more investments to give higher yields and better food security. FI assists in providing them with insurance to defend their assets from external shocks. SDG-5 covers gender equality, and it is also entwined with FI, as it will result in women's social-economic development. This will reduce their risk of exploitation in the informal sector and enable them to engage in productive economic activities. With financial constraints and the inability to keep collateral, women often cannot procure loans [25]; and FI will assist in potential financial development possibilities [26]. This will improve household well-being and enable them to invest in the health and education of their kids, too [27].

SDG-8 promotes long-term, inclusive, and sustainable economic development; full and productive employment; and decent work for all people, regardless of their background. Therefore, the formal financial institutions around the world are taking many significant

steps to provide full finance to the needy, small entrepreneurs and those unbanked. Micro-finance institutions (MFIs) have been set up and helped by many development agencies all over the world so that these customers who are not banked can get financial help [28]. MFIs have contributed significantly to the development of a self-sustaining financial system for the poor and increased entrepreneurial talent [29] and socio-economic development [30–35]. SDG-8 focuses on fostering sustainable economic growth and full and productive employment, and SDG-9 focuses on supporting innovation and sustainable industrialization.

The fundamental purpose of the current research is to examine the prevailing research on FI and sustainable growth and suggest answers to the following research questions:

RQ1: What are the significant themes of research in this domain?
RQ2: Which drivers influence more in achieving financial inclusiveness?
RQ3: How can drivers of FI with the mediation of financial literacy influence sustainable growth?
RQ4: How are financial initiatives related with sustainable growth?

To find answers to these pertinent questions, the present research was undertaken. Using a survey technique with inputs from customers using bank services, the study examines the major drivers of FI. It attempts to understand how drivers of FI through the mediation of financial literacy (FL) influence sustainable growth. It also attempts to investigate the financial initiative's direct impact on sustainable growth. The study uses a Partial Least Squares-Structural Equation Modeling (PLS-SEM) technique to relate drivers of FI, FL, and financial initiative with sustainable growth measured through the achievement of the SDGs.

The related research objectives are:

O1: To identify the impact of the drivers of FI on sustainable growth.
O2: To analyze financial literacy's mediation effect between the FI drivers and sustainable growth.
O3: To investigate the impact of financial initiatives on sustainable growth.
O4: To design a model relating the drivers of FI, financial literacy, and financial initiatives with sustainable growth.

Section 1 introduces the concept of FI, financial literacy, and financial initiatives on sustainable growth. Based on the need for the study, it raises the research questions. Section 2 examines FI from the perspective of the drivers of FI, such as technology, usage, and digitalization. This section also reviews the financial initiatives covering financial programs and policy. Section 3 highlights the research design and methods used to achieve the objectives. Section 4 presents the measurement and structural model. Two control variables were used, and the results are reflected through the two models; the second model is with the control variables. The study designs a PLS-SEM model to examine the impact of the FI drivers through financial literacy and financial initiatives on sustainable growth. Section 5 covers the discussion and conclusions section, reporting the new findings and a comparison with research in a similar area. The last section suggests the implications, limitations, and areas for future research.

## 2. Review of Literature and Hypothesis Development

The study covers a comprehensive review to lay the foundation for the conceptual model. The review of the literature in the current research has been classified under the following headings:

### 2.1. Drivers of FI
#### 2.1.1. Usage

Swamy [36] applauded the FI efforts of India's government, especially from 1991 to 2005, to make banks reach out to rural areas. Bassant [37] highlighted that for achieving growth with equality, commercial banks must opt for cost-effective technology, such as zero-balance bank accounts, point of sale, mobile banking, and ATMs. Consequently, Camara and Tuesta [38] covered three dimensions of FI: usage, access, and barriers. Usage covers having a financial product, a savings account, and a loan. Access covered the

approachability of ATMs, the no. of bank branches, and financial products and services. Barriers included affordability, documentation, distance, and trust. Gine and Townsend [39] revealed a positive linkage between economic development and geographic outreach. Beck et al. [40] considered outreach through access and usage dimensions, and they concluded that usage plays the most prominent role and enables customers to facilitate payments through a debit card and through a savings account, and it allows for asset purchasing, owning a home, educating children, and also to maintain reserves for retirement. Allen et al. [41] cogitate FI through the usage of formal deposit accounts. A stream of thought has focused on the usage of and access to formal financial services [42–44]. In light of these, it is pertinent to consider usage in the present study. Therefore, the related hypothesis is:

**H1a.** *The usage indicator is positively associated with financial inclusion.*

### 2.1.2. Digitalization

The introduction of information and communication technologies (ICTs) and m-banking has given a new face to digitalization [45,46]. Similarly, Demombynes and Thegeya [47] concluded that m-banking with the latest financial services helped transform the lives of the Kenyan population. Many countries have initiated digitalization through ICTs to provide fast, cheap, and accessible financial services. There are many examples of countries using ICT as a medium like mobile money: CELPAY in Zambia; M-PESA in Kenya; and WIZZIT in South Africa. In India, we have the facility of cash transfer through (UIADI) Aadhar and the Unified Payments Interface (UPI). Thus, it is evident that digitalization is an essential driver of FI. GPFI [48] reported that digitalization encourages the user to access digital services and financial products efficiently. The ease of access through digitalization will remove the barriers to FI. Ghosh [49] has reaffirmed that the (Adhar) biometric identification system, with its linkage to bank accounts and other financial services, has a positive influence on FI.

Similarly, Onaolapo [50] suggested that FI can be delivered smoothly in the country through information and communication technology (ICT). Thus, the literature indicates that digitalization is playing an essential role in establishing a financial network in society. Financial technology, including digital payments and mobile money accounts, has helped boost FI [51,52]. Therefore, we have taken digitalization as one of the drivers of FI. Hence, we hypothesize:

**H1b.** *Digitalization is positively associated with financial inclusion.*

### 2.1.3. FinTech

Financial Technology (FinTech) is the new technology to improve and automate the delivery and use of financial services. The first wave of FinTech ushered in innovation across all phases of the customer life cycle; however, the reach was limited to the affluent sections of society. Thus, it becomes evident that without considering FinTech as a driver of FI, the research may not be complete. Point-of-sale devices and networks communicate between the post office agent, retail agent, and financial service provider. Fintech, along with fund transfer and the payment of bills, also facilitates online trading and mutual fund investment [53]. Though massive efforts are being taken to push digital payments, the picture is rather gloomy as only 2% of merchants enabled point-of-sale-based cashless payments [54]. Thus, as technology changes very fast, it was thought to understand from the customer's perspective how relevant FinTech was in inducing a change in FI. As the target population approached a rural segment too, it was pertinent to include their opinion and draw a unified perception of urban and rural customers.

Moreover, the focus of FinTech is changing from facilitating e-payments or transactions to building a relationship. Based on these views, in the current study, it was considered a separate driver of FI and digitalization was taken to have customers' perceptions regarding digital financial services. Kass-Hanna et al. [55] suggest that national FI strategies continue to lean toward digital finance with the FinTech movement.

Therefore, we hypothesize that:

**H1c.** *FinTech is positively associated with financial inclusion.*

Thus, the first hypothesis is:

**H1.** *Usage, digitalization, and FinTech are positively associated with FI.*

After reviewing the drivers of financial inclusion, the following section deals with financial literacy.

### 2.2. Review of Financial Literacy

Financial literacy (FL) enables financial planning and also assists in making effective financial decisions [56]. In view of Lusardi and Mitchell [57], financially sound people were more effective in financial planning and debt management. Lusardi et al. [12] opined that financially literate individuals have better knowledge about how to generate, spend, invest, and save money. Similarly, Grohmann et al. [58] related the expansion of bank branches in rural and urban areas to be associated with improved financial literacy and enhanced FI. Researchers across the globe believe that FI can be achieved through financial competencies by improving financial literacy. However, Atkinson and Messy [59] considered a low level of financial skill and knowledge as the major reason for lower levels of FI in any economy. They recommended that policymakers induce banks and financial institutions to conduct training programs to improve the FI level. Ramakrishna and Trivedi [60] recognized that technology positively influences FI. This was also reverberated by Rastogi and Ragabiruntha [61]. Innovation and technology through literacy can intensify FI, because it can circumvent prevailing structural and infrastructural challenges and directly reach the needy ones [62]. Thus, that is the reason we have taken financial literacy as a mediating variable. Okello et al. [63] have also used financial literacy as a mediator between social networks and FI. Both the direct and indirect effects of FL with FI emerge as significant, which indicates the important role played by FL in FI. Taking this as a pointer for future research, we want to examine the mediation effect of FL between the drivers and sustainable growth in this study. The drivers of FI with the mediation of FL should lead to sustainable growth. Hence, we hypothesize that:

**H2.** *Financial literacy mediates between the drivers of financial inclusion and sustainable growth.*

Next, the research examines the relation of the financial initiative on sustainable growth.

### 2.3. Financial Initiatives

The financial initiatives may play a critical part in the development of FI by allowing the nation to be financially accessible to all people. In the study, financial schemes and policies have been examined to provide enabling environments that are financially well sound. The literature related with financial policy and financial schemes has been presented in this section.

#### 2.3.1. Financial Schemes

Many national and international institutions are leading major policy initiatives and schemes to bridge the gap between FI. Around 35 countries have adopted a National Financial Inclusion Strategy (NFIS) to accelerate sustainable growth. Some countries have modified and restructured their NFIS [64]. In India, major steps have been initiated by RBI in Basil-III norms. Along with increased regulations and supervision of financial institutions, there is a need for the expansion of bank branches in unbanked/rural areas. Policy changes are being introduced for safer banking, risk management, and for accelerating liquidity [65].

Moreover, Italy is an example where poverty levels are reduced through various schemes [66]. Other schemes related with easy access to financial services and zero-balance savings accounts offered by the Nepal government to female heads of households led

to around 84% of women opening their B/As [67]. Similarly, the Indian government has initiated several programs like Pradhan Mantri Jan Dhan Yojna. Therefore, we have analyzed whether financial initiatives taken in India have been helpful in achieving FI and sustainable growth.

Kaboski and Townsend [68] indicated that the Thailand government has taken the initiative to provide micro-credit loans to rural areas by introducing the "Village Fund Program". The Reserve Bank of India, in 2006, permitted banks to use intermediaries as business facilitators (BFs) or business correspondents (BCs) for delivering financial/banking services. Joshi [69] has highlighted a significant role played by financial intermediaries in FI. As indicated by Dugyala [70], reinforcing the initiatives of financial intermediaries such as microfinance institutions and banks is needed. RBI initiated to encourage savings for the Chiller bank program in 2015 to encourage children to open and operate savings bank accounts independently.

### 2.3.2. Financial Policy

FI has been widely accepted as a goal for the financial sector and economic growth during the last several years by policymakers throughout the world. Cohen [71] opined that government and financial institutions should make effective policies, especially on FL in rural and urban areas, for financial intermediaries' involvement. The financial intermediaries and banking channels can deliver financial literacy programs effectively [72]. The Reserve Bank of India focuses on unique programs and policies to successfully achieve FI in the country. It employs a bank-led approach, such as Basic Savings Bank Deposits (BDSD) accounts for the economically disadvantaged, simple Know Your Client (KYC) norms, and directions to open more bank branches in rural areas. The common service centers (CSC) have been set up in rural areas, providing electronic commercial services and e-governance to rural residents. Therefore, financial policies play an essential role in attaining FI and fostering sustainable growth.

The related hypotheses are:

**H3.** *Financial schemes and financial policy have a positive relation and are sub-dimensions of financial initiatives.*

**H4.** *There is a positive relation between financial initiatives and sustainable growth.*

The current research has used sustainable growth measured through SDGs 1, 3, 5, 8, 9, 10, 11, and 17 as a dependent variable. Thus, examining the existing literature on sustainable development goals and how the drivers of FI, financial literacy and financial initiatives are related to sustainable growth is mandatory.

### 2.4. Sustainable Growth

The basic purpose of any economy is to have sustainable growth, which offers basic financial services to unbanked and rural areas and reduces disparities. SDG-1 focuses on eliminating extreme poverty. It also states that the poor and the vulnerable should have equal rights to access financial services, including microfinance. Similarly, SDG-5 is about promoting gender equality. Access to financial services, such as credit, helps women assert their economic power [25]. We would also like to refer to SDG-9, promoting innovation and sustainable industrialization. Sustainable growth advocates equitable opportunities for people during economic growth. It ensures benefits for all income groups.

Examining the researchers' perspectives on economic development in sustainable and inclusive growth is necessary. McKinnon and Shaw [73] concluded that expanding bank branches in rural/urban areas positively affects economic growth. Levine [74] and Beck et al. [40] also found a well-established financial system to be positively linked with the economy's growth. Khan [75] supported that a well-defined financial system encourages investment and promotes growth. Indeed, Bertram et al. [76] concluded that FI served as a prerequisite for inclusive economic development in Nigeria. Hariharan

and Marktanner [77] supported the impact of FI economic growth and development as they observed a high positive correlation between FI and total factor productivity (TFP), which translates to growth. The same thoughts were reverberated by Kim et al. [78] in their research of 55 member countries of the Organization of Islamic Cooperation (OIC), where a positive relation of FI was observed with economic growth. Park and Mercado [79] found FI to be positively correlated with per capita income. Ibor et al. [80], in a study on Bangladesh, concluded that financial inclusiveness has helped in alleviating poverty and an improvement of living standards.

However, Zins and Weill [81] used a probit model on 37 African countries and found that educated, richer, and older individuals are more financially included. Access to formal financial services in an economy provide new and equal opportunity for investment for individuals/businessmen [82]. Increased FI improved indicators such as income, the standard of living, health, education, and poverty reduction [83]. Thus, it becomes essential to find out how FI drivers and financial initiatives have helped in achieving sustainable growth. Sustainable growth has been measured by the customer's perception regarding how FI helps in achieving dimensions covering aspects from reducing inequalities and enhancing health to fostering growth and innovation through SDGs 1, 3, 4, 5, 8, 9, 10, 11, and 17. SDGs were adopted in 2015 by the United Nations (UN) with the aim of ending human poverty in all of its forms in the world. Access to formal financing assists in achieving broader goals, such as ending poverty (SDG-1), improving health and education (SDGs 3 and 4), reducing gender inequality (SDG-5); improving entrepreneurial activity and innovation and growth (SDGs 8 and 9) [84–86]. SDG-10 is about reducing inequality SDG-11; making cities and other places where people live safe, resilient, and sustainable is related to SDG-17, i.e., reinvigorating the global cooperation for sustainable development by strengthening the implementation mechanisms. The study will also be able to focus on which SDG has a higher loading in sustainable growth as per the customer's perception. There are SLR studies covering FI and SDGs; however, such a study covering a survey-based analysis has not been undertaken. Thus, the related hypothesis is:

**H5.** *Sustainable growth is measured through the consumer's perception of how FI helps in achieving dimensions covering aspects from reducing inequalities and enhancing health to fostering growth and innovation through various SDGs, viz., SDGs 1, 3, 5, 8, 9, 10, 11, and 17.*

**H6.** *Drivers of FI with the mediation of financial literacy and financial initiatives positively influence sustainable growth.*

## 3. Research Design and Methodology

Research design and methodology section covers research framework, data, research methodology, and operationalization.

### 3.1. Research Framework

There are no synergies regarding FI drivers, financial literacy, financial initiative, and sustainable growth. The outcomes of drivers of FI and their impact on sustainable growth vary significantly, which was the prime reason for undertaking the current study. We theorize that drivers of FI are positively related to sustainable growth. This relation is strengthened through the mediation effect of financial literacy. We also theorize that there is a positive relation between financial initiatives and sustainable growth. The research framework is presented in Figure 1.

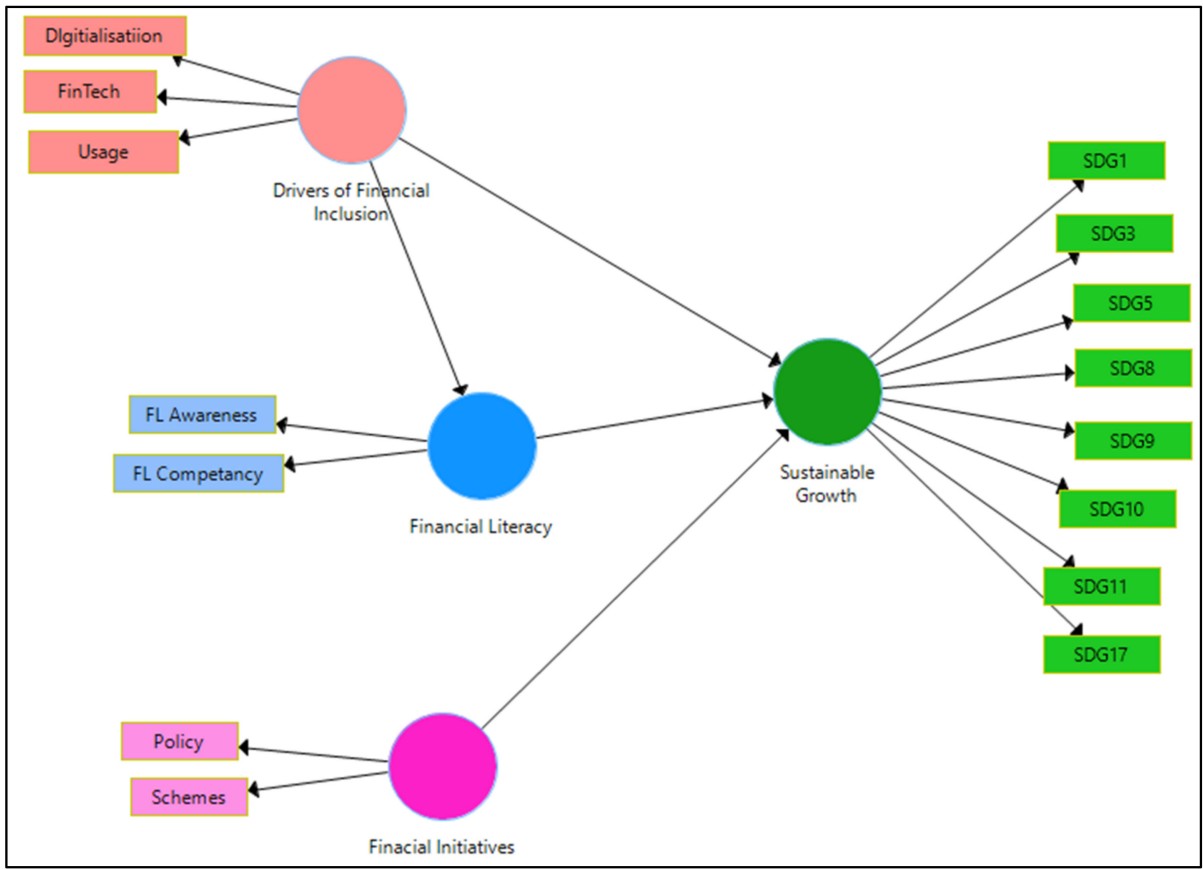

**Figure 1.** Research framework. Source: Author's creation.

*3.2. Data*

A cross-sectional design was used for this research and data were collected through a structured questionnaire from customers. The data collection timeline was from August 2019 to October 2020. The population for this research was drawn from customers from different north Indian states, which are Haryana, Punjab, Himachal Pradesh, New Delhi, Chandigarh, and Uttarakhand. The study used a 5-point Likert scale survey. To ensure the questionnaire's content validity, it was first delivered to a convenience sample of 90 persons. Academicians and business professionals were included in this pilot group. The feedback from the pilot group was used to improve the questionnaire. The pilot group also recommended adding a few items in drivers of FI to cover developing nations still in their development stage. We distributed 1993 surveys and received 1325 replies, resulting in a response rate of 66.4 percent. To represent the overall population, the sample included both urban and rural locations, genders, graduates and postgraduates, service class persons, and self-employed people. The attempt was made via revisits to increase sample participation. This was possible after the researcher personally visited banks to collect customer data, and a third party was not employed.

Table 1 summarizes the characteristics of the customers surveyed. Out of the total 1325 users, 51% were males and 49% were females. Among the respondents, 37% were from rural and 63% from urban sectors. Regarding age group, the people above 51 years were less. The majority of respondents were from private sector banks. There was a dominance of urban respondents in the sample. However, the sample is a representative sample as per the North India statistics, where there is a dominance of the urban and male population.

**Table 1.** Demographic detail.

|  | Number of Respondents | Valid Percentage |
|---|---|---|
| Gender |  |  |
| Male | 676 | 64.7% |
| Female | 649 | 35.3% |
| Age |  |  |
| Less than 35 | 505 | 38.11% |
| 35 to 50 | 555 | 41.8% |
| 51 and above | 265 | 20% |
| Educational Qualification |  |  |
| Undergraduates | 284 | 21.43% |
| Graduates | 602 | 45.43% |
| Postgraduates | 439 | 33.13% |
| Category of bank |  |  |
| Public sector | 452 | 34.11% |
| Private sector | 735 | 55.47% |
| Small finance institution | 138 | 10.41% |
| Region |  |  |
| Rural | 490 | 36.98% |
| Urban | 835 | 63.01% |

Source: Self-calculated through SPSS.

### 3.3. Methodology

The current research has used the variance-based Partial Least Squares-Structural Equation Modeling (PLS-SEM). It is a multivariate analysis method based on a series of ordinary least squares regressions and has higher levels of statistical power than covariance-based SPSS-AMOS [87]. The study uses Smart PLS 3.2.0 [88] to compute the path model. The further bootstrapping technique has been used to examine the loadings' significance. The next section discusses the results. Initial results are based on factor analysis. This is followed by a model designed using PLS-SEM

### 3.4. Operationalization

The study used structured questionnaires to collect data from respondents. The survey was conducted in three rounds from August 2019 to December 2019, February 2020 to August 2020, and November 2020 to December 2021. To see whether there is a nonresponse bias, the mean differences in critical variables across early (n = 714) and late respondents (n = 611) were tested. There were no significant changes between the two samples, which means there was no non-response bias.

Normal distribution plots, skewness, and kurtosis have been used to evaluate the assumption of normal distribution (Table 2). For the normal distribution, the skewness should be near zero and a negative value indicates skewness toward the left. Similarly, the kurtosis values are less than 3; thus, the data fulfill the criteria for normal distribution.

**Table 2.** Sample characteristics.

|  | Mean | Standard Deviation | Kurtosis | Skewness |
|---|---|---|---|---|
| Drivers of FI |  |  |  |  |
| Usage | 4.193 | 0.391 | −0.549 | −0.410 |
| Digitalization | 4.124 | 0.451 | 0.725 | −0.434 |
| FinTech | 3.850 | 0.511 | 1.604 | −0.492 |
| Financial Literacy |  |  |  |  |

**Table 2.** *Cont.*

| | Mean | Standard Deviation | Kurtosis | Skewness |
|---|---|---|---|---|
| FL Awareness | 4.150 | 0.471 | −0.272 | −0.201 |
| FL Competency | 3.805 | 0.430 | −0.617 | −0.345 |
| Financial Initiatives | | | | |
| Financial Schemes | 4.102 | 0.435 | −0.718 | −0.240 |
| Financial Policies | 3.978 | 0.430 | −0.617 | −0.345 |
| Sustainable Growth (SDG) | | | | |
| SDG1 | 4.137 | 0.460 | 1.199 | −0.425 |
| SDG3 | 4.152 | 0.491 | −0.424 | −0.463 |
| SDG5 | 3.838 | 0.732 | −0.431 | −0.453 |
| SDG8 | 4.146 | 0.438 | 0.001 | −0.671 |
| SDG9 | 4.076 | 0.674 | 1.120 | −0.405 |
| SDG10 | 4.121 | 0.560 | −0.120 | −0.470 |
| SDG11 | 3.542 | 0.438 | −0.113 | −0.410 |
| SDG17 | 4.187 | 0.499 | −0.165 | −0.591 |

Source: Self-calculated through SPSS.

## 4. Data Analysis and Results

The data analysis process is divided into two sections. The first confirms the factor structure of the measurement items of the drivers of FI, financial literacy, financial initiatives, and sustainable growth. The second stage investigates the relative importance of FI, financial literacy, and financial initiatives in explaining sustainable development. The measurement model helps to decide the properties of the scales and the structural model to establish the relationships among the variables.

### 4.1. Measurement Model

The results are represented through a measurement model to check the reliability and for validation in Section 4.1. This is followed by the structural model highlighting the results in Section 4.2. The measurement model could be examined through construct reliability, convergent validity, and discriminant validity.

As depicted in Table 3, the composite reliability (CR) values are more significant than the recommended threshold criterion of 0.70 [89]. The Cronbach alpha value for all constructs is between 0.770 and 0.893. The composite reliability values have a range of 0.881 to 0.948 (Table 3). This highlights that the construct validity and the reliability of the model are good and acceptable. According to Fornell and Larcker [90], the convergent validity of the constructs is examined by factor loadings and the average variance extracted (AVE). The value of the factor loadings and average variance extracted (AVE) should exceed the minimum requirement of 0.50 [91] for the explained variance to be greater than the measurement error. In the current study, the resulting value of the factor loadings is 0.611 to 0.914, and the AVE lies between 0.502 and 0.813. This condition is also satisfied. The indicators in the reflective measurement model show satisfactory levels of indicator reliability. As shown in Table 3, the outer loadings are greater than 0.70 for most of the items. However, in the case of SDG3, the value of the factor loading is 0.644; for SDG 5, it is 0.648; and for SGD 9, it is 0.611. As these are important for research, few researchers have suggested retaining the items if the values are greater than 0.60. Hence, we have retained them for further analysis.

The average variance extracted (AVE) greater than 0.50 supports the measures' convergent validity. The discriminant validity [90] was measured by comparing the values of the square root of AVE. It is recommended that the value of the square root of AVE should be larger than the inter-construct correlations (Table 4). The results confirm that the reflective constructs exhibit discriminant validity.

**Table 3.** Measurement model.

| | Factor Loadings | Cronbach's Alpha | Rho_A | Composite Reliability | Average Variance Extracted (AVE) |
|---|---|---|---|---|---|
| Sustainable Growth | | 0.847 | 0.857 | 0.881 | 0.502 |
| SDG1 | 0.711 | | | | |
| SDG3 | 0.644 | | | | |
| SDG5 | 0.648 | | | | |
| SDG8 | 0.758 | | | | |
| SDG9 | 0.611 | | | | |
| SDG10 | 0.725 | | | | |
| SDG11 | 0.712 | | | | |
| SDG17 | 0.740 | | | | |
| Drivers of FI | | 0.831 | 0.903 | 0.895 | 0.740 |
| Usage | 0.893 | | | | |
| Digitalization | 0.878 | | | | |
| FinTech | 0.840 | | | | |
| Financial Literacy | | 0.893 | 0.884 | 0.948 | 0.730 |
| FL Awareness | | | | | |
| FL Competency | | | | | |
| Financial Initiatives | | 0.770 | 0.776 | 0.897 | 0.813 |
| Schemes | 0.914 | | | | |
| Policies | 0.889 | | | | |

Source: Self-calculated through PLS-SEM.

**Table 4.** Heterotrait-monotrait ratio (HTMT).

| | Sustainable Growth | Drivers of FI | Financial Initiatives | Financial Literacy |
|---|---|---|---|---|
| Sustainable Growth | | | | |
| Drivers of FI | 0.842 | | | |
| Financial Initiatives | 0.840 | 0.847 | | |
| Financial Literacy | 0.683 | 0.816 | 0.831 | |

Source: Self-calculated through PLS-SEM.

The next step was to check the outer and inner variance inflation factor (VIF). The VIF values are presented (Table 5). As highlighted, the outer and Inner VIF values are less than 3 and in the acceptable range [92]. Thus, the collinearity is low, as indicated by a VIF value lower than 3; thus, no indicator was removed.

### 4.2. Structural Model

The results of the measurement model highlight that the construct reliability, convergent validity, and discriminant validity are all in the acceptable range. When the measurement model had been verified, the relationship dimensions of the model and sustainable growth were performed. The structural model results, as depicted in Figure 2, show that the beta value between the drivers of FI and financial literacy is 0.877 and between financial literacy and sustainable growth is 0.370. The indirect effect is 0.324 (0.877 × 0.370), while the direct effect of the drivers of FI and sustainable growth is 0.152. Further financial initiatives are positively and directly related to sustainable growth, and the beta value

is 0.472. The results indicate that with the mediation of FL, the impact of the drivers on sustainable growth improved and was significant too.

**Table 5.** Outer and Inner VIF.

| Outer VIF | | Inner VIF | | | |
|---|---|---|---|---|---|
| | **VIF** | | **Sustainable Growth (SDG)** | **Drivers of FI** | **Financial Initiatives** | **Financial Literacy** |
| Digitalization | 2.357 | Sustainable Growth | | | | |
| FinTech | 1.923 | Drivers of FI | | | | 1.000 |
| Financial Literacy | 1.000 | Financial Initiatives | 1.442 | | | |
| Policy | 1.646 | Financial Literacy | 1.442 | | | |
| SDG1 | 1.773 | | | | | |
| SDG10 | 1.743 | | | | | |
| SDG11 | 1.553 | | | | | |
| SDG17 | 1.842 | | | | | |
| SDG3 | 1.370 | | | | | |
| SDG5 | 1.359 | | | | | |
| SDG8 | 1.894 | | | | | |
| SDG9 | 1.424 | | | | | |
| Schemes | 1.646 | | | | | |
| Usage | 1.772 | | | | | |

Source: Self-calculated through PLS-SEM.

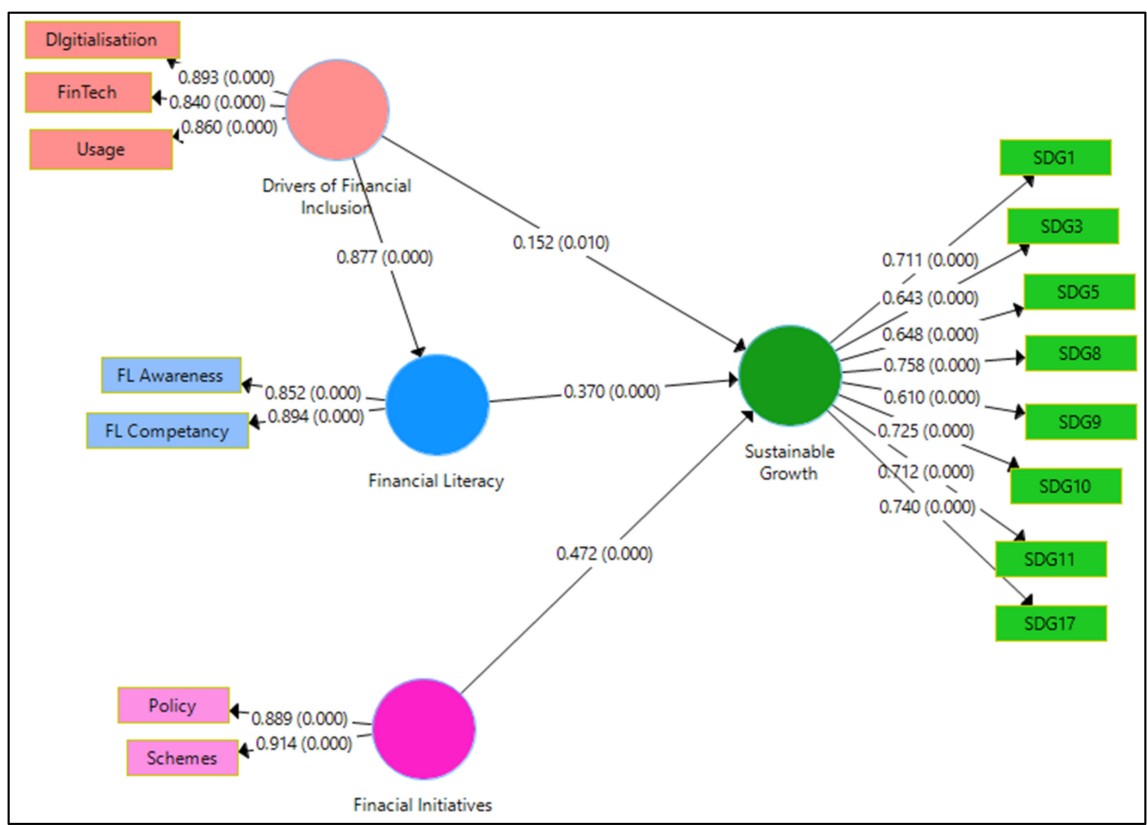

**Figure 2.** PLS-SEM bootstrapping model relating drivers of FI, financial literacy, and financial initiatives with sustainable growth. Source: Author's calculation through the help of PLS-SEM.

Figure 2, along with Table 6, will help understand the status of the hypotheses. The outer loadings of usage are 0.860 and are the highest amongst the drivers of FI. Hence, we accept H1a, that usage is positively associated with FI. The outer loading of digitalization is 0.893; thus, we accept H1b: Digitalization is positively associated with FI. For FinTech, the outer loading of FinTech is 0.840. Thus, H1c: FinTech is positively associated with FI and has also been accepted. Thus, the first hypothesis that H1: Usage, digitalization and

technology are positively associated with FI has been accepted as all the dimensions have high outer loadings.

**Table 6.** Structural model analysis with control variables.

| | Model 1 | | | | | Model 2 (with Control Variables) | | | | |
| --- | --- | --- | --- | --- | --- | --- | --- | --- | --- | --- |
| | Original Sample (O) | Sample Mean (M) | Standard Deviation (STDEV) | T Statistics (\|O/STDEV\|) | *p* Values | Original Sample (O) | Sample Mean (M) | Standard Deviation (STDEV) | T Statistics (\|O/STDEV\|) | *p* Values |
| Drivers of Financial Inclusion -> Financial Literacy | 0.152 | 0.885 | 0.027 | 32.490 | 0.000 *** | 0.152 | 0.160 | 0.059 | 2.555 | 0.011 * |
| Drivers of Financial Inclusion -> Sustainable Growth | 0.877 | 0.484 | 0.046 | 10.421 | 0.000 *** | 0.877 | 0.886 | 0.027 | 32.285 | 0.000 *** |
| Financial Initiatives -> Sustainable Growth | 0.472 | 0.468 | 0.040 | 11.763 | 0.000 *** | 0.472 | 0.468 | 0.040 | 11.788 | 0.000 *** |
| Financial Literacy -> Sustainable Growth | 0.372 | 0.366 | 0.046 | 8.019 | 0.000 *** | 0.372 | 0.371 | 0.048 | 7.673 | 0.000 *** |
| Gender -> Sustainable Growth | | | | | | −0.018 | 0.015 | 0.021 | 0.708 | 0.379 |
| Region -> Sustainable Growth | | | | | | 0.016 | 0.016 | 0.021 | 0.783 | 0.430 |
| | R Square | | | | | R Square Adjusted | | | | |
| Sustainable Growth | 0.786 | | | | | 0.786 | | | | |
| Financial Literacy | 0.769 | | | | | 0.769 | | | | |

Source: Self-calculated *** $p \leq 0.001$; * $p \leq 0.05$.

The next hypothesis was that H2: Financial literacy mediates between the drivers of FI and sustainable growth. Financial awareness and financial competency had outer loadings greater than 0.850. Hence, it can be inferred that financial literacy comprises FL awareness and FL competency. The literature suggests that financial literacy will have a positive impact on sustainable growth. This study tries to analyze whether financial literacy mediates between the drivers of FI and sustainable growth. For this, we need to access the direct path of FI's influence on sustainable growth and the indirect path through financial literacy as a mediator. The results indicate that the FI drivers influence the economy's sustainable growth. The direct path coefficient is 0.152 (t-statistics 32.490) and is significant ($p < 0.001$). The indirect path co-efficient is (0.877 × 0.370) and the t-statistics are also significant ($p < 0.001$). The strength of the relationship has improved with the mediation of financial literacy. *Thus, H2: Financial literacy mediates between drivers of financial inclusion and sustainable growth has been empirically validated.*

The next hypothesis is H3: Financial schemes and financial policy have a positive relation and are sub-dimensions of financial initiatives. As the loadings of both the dimensions, financial policy (0.889) and financial schemes (0.914), are high, we accept H3: Financial schemes and financial policy have a positive relation and are sub-dimensions of financial initiatives. It is now important to examine the relation between financial initiatives and sustainable growth. A beta value of 0.472 and a t-value of 11.763 and ($p < 0.001$) support the acceptance of the hypothesis, viz., *H4: There is a positive relation between financial initiatives and sustainable growth.*

The results of the present study highlight that the drivers of FI, financial literacy, and financial initiatives influence sustainable growth. These three predictors explain

78.6 percent of the variation in sustainable growth. These results indicate that all the predictors considered in the study influenced sustainable growth, although the degree of influence is varied. The results confirm *H5: Sustainable growth is measured through the consumer's perception of how FI helps in achieving dimensions covering aspects from reducing inequalities and enhancing health to fostering growth and innovation through various SDGs, viz., SDGs 1, 3, 5, 8, 9, 10, 11, and 17*, as all outer loadings are high for the undertaken SDGs. The findings highlight that the drivers of FI with the mediation of financial literacy emerge as an important predictor. An important finding is that emerging financial initiatives also significantly impact sustainable growth. This lends support to *H6: Drivers of FI with the mediation of financial literacy and financial initiatives positively influence sustainable growth.*

### 4.3. Structural Model with Control Variables

In the next stage, we introduced the control variables and checked the structural model results again (Figure 3). Region and gender were introduced as the control variables. The results were almost similar. The beta value between financial initiatives and sustainable growth (SDG) was 0.472. The values were significant for relations between the drivers of FI and financial literacy, and between financial literacy and sustainable growth (SDG). The results were also significant for financial initiatives and sustainable growth (SDG). The model also depicts that results were not significant for gender and sustainable growth (SDG) and also for the region and sustainable growth. Furthermore, the beta value for gender is "−0.018" (*p*-value: 0.379), indicating that the results are supportive for males rather than females.

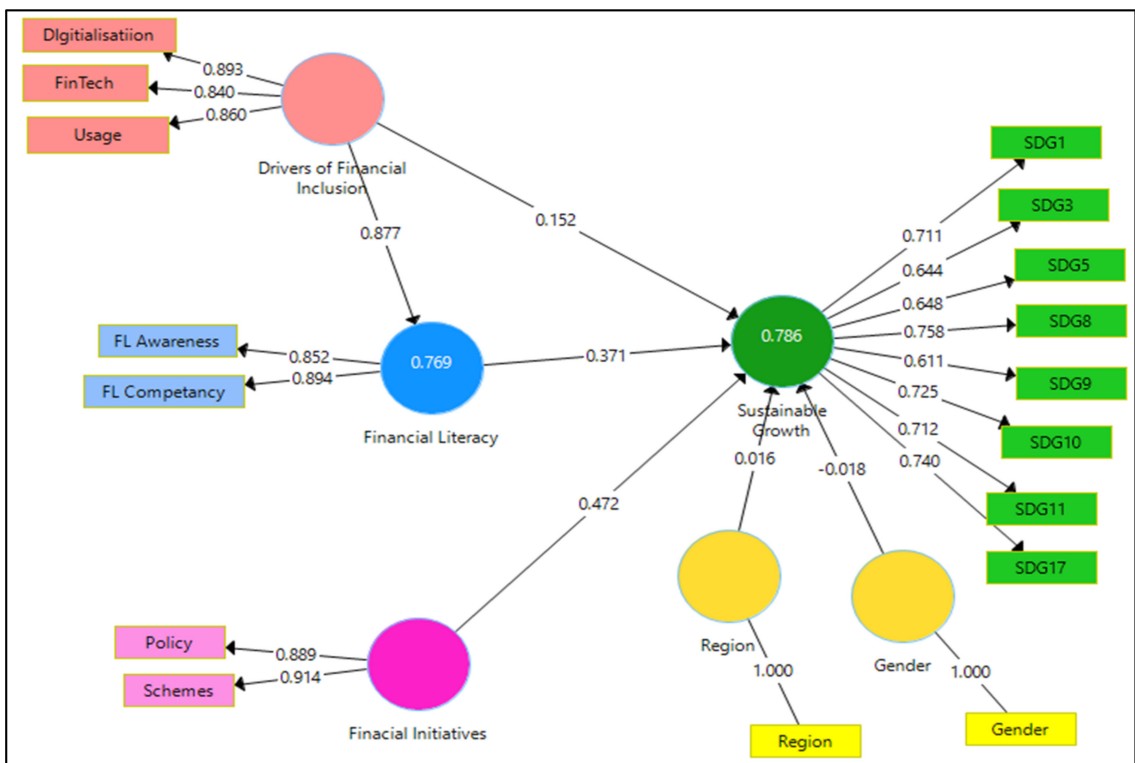

**Figure 3.** PLS-SEM model with a control variable. Source: Author's calculation through the help of PLS-SEM.

Similarly, the beta value for the region is 0.016 (*p*-value: 0.430), indicating a positive relation with the urban rather than rural sector. Women with access to financial services may control personal and have productive expenditures [93]. Thus, we accept *H7: Gender and region are the control variables and do not influence the endogenous variable, viz., sustainable growth*. However, the results of the current study highlight the advantage for males. Thus,

this may be taken as a lacuna and FL may be provided to females to avail advantages of financial inclusiveness and its transmission to sustainable growth.

## 5. Discussion and Conclusions

The aggregative result of the study in terms of the status of hypotheses has been shared in Table 7.

**Table 7.** Status of hypotheses.

| Hypotheses | Status |
| --- | --- |
| *H1: Usage, digitalization, and FinTech are positively associated with FI.* | *Empirically Supported* |
| *H1a: Usage indicator is positively associated with financial inclusion.* | *Empirically Supported* |
| *H1b: Digitalization is positively associated with financial inclusion.* | *Empirically Supported* |
| *H1c: FinTech is positively associated with financial inclusion.* | *Empirically Supported* |
| *H2: Financial literacy mediates between the drivers of financial inclusion and sustainable growth.* | *Empirically Supported* |
| *H3: Financial schemes and financial policy have a positive impact and are sub-dimesions of financial initiatives.* | *Empirically Supported* |
| *H4: There is a positive relation between financial initiatives and sustainable growth.* | *Empirically Supported* |
| *H5: Sustainable growth is measured through consumers' perception of how FI helps in achieving dimensions, covering aspects from reducing inequalities and enhancing health to fostering growth and innovation through various SDGs, viz., SDGs 1, 3, 5, 8, 9, 10, 11, and 17.* | *Empirically Supported* |
| *H6: Drivers of FI with the mediation of financial literacy and financial initiatives positively influence sustainable growth.* | *Empirically Supported* |
| *H7: Gender and region are the control variables and do not influence the endogenous variable, viz., sustainable growth (SDG).* | *Empirically Supported* |

Source: Self-calculated.

The results indicate that all the hypotheses have been accepted. Starting primarily with the drivers of FI, viz., usage, digitalization, and FinTech, the results suggest that these are significant drivers of FI and are positively influencing FI. Bhandari [94] has also highlighted penetration and usage as important dimensions of FI. An earlier study by Gu, Lee, and Suh [95] emphasized trust and usage as important indicators inducing m-banking in emerging economies. The results of the current study emphasize the digitalization indicator emerges as the most important driver, followed by usage and FinTech. The empirical findings of Duncombe and Boateng [96] and Barbu et al. [97] reveal that technological innovations, viz., connectivity, improve access to financial products for the public. This has also been reflected in the current research as FinTech emerges. The thoughts of Kim et al. [78] support the FI-sustainable development goal nexus for the Organization of Islamic Cooperation (OIC) economies and similar results were also reverberated by Sharma [10]. FI is related to sustainable growth [75,98,99]. This is also endorsed by Ryu and Ko [100] suggesting customers' hesitancy to adopt FinTech, suggesting effort is needed to promote it.

Chithra and Selvam [101] supported a positive relation between deposit and credit penetration on FI in India. Financial initiatives help boost FI. The present study highlights a positive relation of financial initiatives on sustainable growth. This has been indicated in earlier studies by Sarma and Pais [102] and Fungáčová and Weill [103]. The present research depicts a holistic picture by relating drivers of FI, financial literacy, and financial initiatives with sustainable growth measured through customers' perceptions regarding the success of FI through the achievement of the SDGs considered in the model. The strategic collaboration of FI and financial education leads to the financial stability of society and the economy [104]. The present study underlines the importance of FI drivers with the mediation that financial literacy enhances sustainable growth.

A strategy toward meaningful FI is needed to unlock the potential for reducing gender inequalities for dynamizing and sustaining growth. The recent works on FI underscore that through access to financial services and products, and the marginalized population can also manage income in a better and more conducive manner [105,106]. This will diminish poverty [107] and enhance economic activity [108]. However, as indicated by Bateman and Chang [109], caution may be followed by undue reliance on a traditional model of MFI. This along with reliance on financial initiatives is essential for sustainable growth. This also underlines the importance of financial literacy, as with literacy, the essence of FI drivers can be achieved, and thus is an important step toward sustainable growth.

Hence, from the above analysis, it can be concluded that drivers of FI, viz., the usage indicator, digitalization, and FinTech, are positively associated with financial inclusion and with the mediation of financial literacy, and they positively influence sustainable growth. Sustainable growth has been measured through customers' perceptions regarding the success of FI through the achievement of selected SDGs, viz., SDGs 1, 3, 5, 8, 9, 10, 11, and 17. Further, it can be concluded that there is also a positive relation between financial initiatives and sustainable growth. The study has added importance as it considers gender and region as control variables and creates a model taking all predictors along with the control variables.

## 6. Implications of the Study

The empirical findings of the present study specify valuable implications for practitioners. Understanding the constructs in the proposed research model is crucial for promoting financial inclusiveness for bankers in India and bankers in emerging economies. To enhance financial inclusiveness and its transmission to sustainable growth, there is a need to continue informing customers about changes in digitalization and FinTech. This study examines the impact of drivers on sustainable growth through the mediation of financial literacy on sustainable growth. The research has empirically corroborated the significant and positive impact of the drivers of FI with the mediation of FL for achieving sustainable growth as measured through the impact on various SDGs. In addition, the study has a rich contribution as it depicts a positive effect of financial initiatives on sustainable growth. This will help other economies to have proper initiatives for enhancing growth through financial initiatives focusing on financial policy and schemes. The results also highlight the importance of using the mentioned FI drivers, financial literacy, and financial initiatives for the achievement of financial inclusiveness success and sustainable growth.

The major purpose of the research was to assess the impact of FI on sustainable growth. Sustainable growth is measured by asking for customers' perceptions about FI success through the achievement of the mentioned SDGs. This study moves beyond the systematic literature covering FI and SDGs and empirically validates the relevance of FI for attaining sustainable growth. The results reflect that customers considered that FI helped in achieving sustainable growth with respect to SDG-8, i.e., improve entrepreneurial activity and innovation and growth, which had the highest loading, followed by SDG-17, to strengthen the means of implementation and revitalizing global partnership for the sustainable development goal. SDG-8, reducing inequalities, was the next priority for consumers. The results were also good for SDG-1: ending poverty. However, there is a need to improve in terms of SDG-3, improving health and education, and SDG-5, reducing gender inequality.

Further, there is a need to focus on the drivers of FI to enhance the success of FI. These implications will help to highlight the interdependence of the drivers of FI and financial literacy for achieving sustainable growth. The relation, as highlighted through the findings, supports the impact on sustainable growth. Thus, segregated policy needs to be intertwined with a dose of financial literacy to enhance financial inclusiveness and sustainable growth.

## 7. Limitations and Future Research

Like any survey-based study, the present research also has some limits and confines. The first constraint has to do with the results' generalizability. As the current research was conducted in North India, validating this research in other Asian countries could test the precision of the findings. This would also be helpful in examining the relation from different cultural perspectives. Secondly, there could be a few drivers which we may have missed. Our endogenous variable of sustainable growth is based on the certain SDGs considered in the model. Earlier FI- and SGD-based research papers focused on secondary data. Moreover, because of time constraints, the study considered survey-based results only. The study has taken gender and region as the control variables. This offers a lot of space for academics to consider the moderating role and experience.

**Author Contributions:** Conceptualization, A.P.; Formal analysis, A.P. and R.K.; Methodology, R.K.; Resources, A.P.; Software, R.K.; Validation, A.P. and R.K.S.; Writing—original draft, R.K.; Writing—review & editing, A.P. and R.K.S. All authors have read and agreed to the published version of the manuscript.

**Funding:** This research received no external funding.

**Institutional Review Board Statement:** Written consent was taken from all the respondents before completing the questionnaire.

**Informed Consent Statement:** Not applicable.

**Data Availability Statement:** All the primary data used for findings in the study are available from corresponding authors upon request.

**Acknowledgments:** We appreciate the editor's recommendations and remarks.

**Conflicts of Interest:** The authors declare that they have no conflict of interest.

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
