# Peer review of "Investigating the Impact of Financial Inclusion Drivers, Financial Literacy and Financial Initiatives in Fostering Sustainable Growth in North India"

_sustainability, doi:10.3390/su141711061_

Round 1

Reviewer 1 Report

Congratulations on your paper! I liked the proposed model and structure of the paper. However there are some minor suggestions:

- page 8. You do not present the hypotheses again once introduced in the text. To be deleted

- Keep only one our of figure 2,3 and 4. I recommend just to keep figure 4 to avoid redundancy

- Section 5 should be only about discussion. Improve the discussion by elaboration more on your findings. How fintech can contribute to financial inclusion? What can be done to increase financial literacy? What policies work best for those financial excluded? are just possible topics. Suggested possible references:

Barbu, C.; Florea, D.; Dabija, D.-C.; Barbu, M. Customer Experience in Fintech. J. Theor. Appl. Electron. Commer. Res. 2021, 16, 1415–1433

Gomber, P.; Kauffman, R.J.; Parker, C.; Weber, B.W. On the Fintech Revolution: Interpreting the Forces of Innovation, Disruption, and Transformation in Financial Services. J. Manag. Inf. Syst. 2018, 35, 220–265.

Kozup, J., and Jeanne Hogarth (2008), Financial Literacy, Public Policy, and Consumers’ Self-Protection—More Questions, Fewer Answers. Journal of Consumer Affairs, 42 (Summer): 127–136.

Ryu, H.-S.; Ko, K.S. Sustainable Development of Fintech: Focused on Uncertainty and Perceived Quality Issues. Sustainability 2020, 12, 7669.

- Place the Conclussions at the end of your paper. Only one or two paragraphs to highlight the essence of your paper. 

Author Response

Reviewers comments and suggestions for manuscript is attached.  Please see the attachment.

Reviewer 2 Report

Dear Authors,

This article presents an interesting study that examines the impact of factors on sustainable growth through financial literacy. The authors set out to understand whether intermediation helps increase the impact of FI drivers on sustainable development. The title of the paper is of great interest to general and specialist readers. However, before introducing the article, we would recommend a few:

1.       The introduction is somewhat larger in scope. It would be useful to split or shorten it and move some parts to the "Hypothesis Formation" section in Chapter 2.

2.       The literature review and hypothesis formation is pretty extensive. We would just like to draw attention to the headings that appear at the end of the pages. Move them to a new page. For heading 2.5 Conceptual Model: colons are shown and are not elsewhere. Alternatively, remove them here or add them to the other chapters. For Figure 1, align the format to align with the text.

3.       Chapter title 3. Measurement Development move to a new page.

4.       In Chapter 4, write at least one or two sentences between the chapter and subchapter. Also align Figure 2 with the text. Same as Figure 3. Add at least one sentence between Figures 2 and 3. Adjust Table 6 to be the same size as the previous tables. Alternatively, if it would be possible to split the table, split it. Edit Figure 4 as the previous figures.

The abstract is very nice. It captures the essence of the article and contains everything. The article is factual. Once the article is resolved and edited, we recommend the article for publication.

Author Response

Thank you for the positive response.  Reviewers comments and suggestions for manuscript is attached.

Reviewer 3 Report

The assertions of the paper seem plausible and the verification through statistical tools does not bring more insight. Further several cases of stochastic dependence are expected between variables of the model. It seems that this is not included in the construction of the model. Further more there are several mistakes to be fixed:

1. On p. 1, l. 34, the work Demirguc-Kunt and Klapper 2013 is not found in the references.

2. On p. 4, l. 178, instead of 'digitization' probably is meant 'digitalization'. This term appears later as digitalisation.

Author Response

(The authors gave the same response as above.)

Reviewer 4 Report

Introduction

The introduction was prepared correctly The authors accurately identified the research gap, justified the choice of the research topic, identified the study objectives and research questions.

Literature review

Although the authors have reviewed the literature quite diligently, I am concerned that the authors have incorrectly formulated the research hypotheses H1 a,b, c and H3. The Hypotheses should specify the assumed direction of change based on the literature review. What does “important” mean in a hypothesis? Is it statistical or general importance? Authors should indicate whether they assume positive or negative impact.(....). I strongly recommend these changes because the research method used fully entitles the authors to formulate such hypotheses. I also suggest that the authors indicate "the impact" in the title instead of "the role"

Title

The authors could consider rewording the title of the article to simplify it a bit.  E.g.  

Determinants of financial inclusion, financial literacy and financial initiatives in sustainable development of north India 

On page 3, the authors should also describe in a bit more detail the critical research on the effectiveness of microfinance institutions (MPIs). These elements should also be included in the discussion. It should be also emphasized, that the financialization of societies also carries many risks for sustainable development.  I ask that the authors take a look at the following examples of research.

- Gosh 2013. Microfinance and the challenge of financial inclusion for development

- Bateman, M. 2010. Why Doesn’t MicrofinanceWork? The Destructive Rise of Neoliberalism, London, Zed Books.

- Bateman, M. and Chang, H.-J. 2012. Microfinance and the illusion of development: from hubris to nemesis in thirty years, World Economic Review, vol. 1, 13–36.

Methodology section

The authors correctly presented the methodological part. However, Due to the research method used, my suggestion is that the authors consider the order of the information presented. First, they more thoroughly justified in the method part the choice of research method and model, and as a separate sub-item included (operationalization (Scales employed) described the operationalization of variables and the adopted scales of variables. Thus: point 3 Method should be divided into a few sections. 3.1. Research framework (model presentation) 3.2. Data,3.3. Methodology, 3.4. Operationalization (scales employed). Such a layout would increase the readability of the article. 

Results

The authors correctly presented the results.

Other 

The authors could consider proofreading - For example, this sentence is a tautology.

" Financial literacy assists 56 in the acquisition of skills essential for financial literacy.." (see p. 2).

Author Response

(The authors gave the same response as above.)

Round 2

Reviewer 3 Report

In spite of the remarkable efforts by the authors the paper is still confined in a qualitative approach of financial inclusion, literacy and initiatives.  This means that the positive association of some sizes is not enough description of the relation between them. The estimation of their dependence can bring new insight in calculation of the financial status. Therefore I have to insist in improvement of the quantitative side of the topic. Furthermore, there are several little defects that should be fixed, as for example

1. The formulation of the hypothesis H1, H2, etc, are repeated several times in the paper. It is enough to make reference to the first appearance. In general the text should be more concise.

2. On p. 10, l. 429, the spelling of the word 'reliability' is not correct.

3. On p. 11, l. 437, instead '0.881' should be '0.893'.

4. On p. 11, l. 438, instead 'Table 2' should be 'Table 3'.

5. On p. 11, l. 443, instead '0.504 to  0.887' should be '0.611 to 0.914'.

6. On p. 11, l. 446, the correction 'Table'.

7. On p. 11, l. 447, instead '0.665; for SDG 5, it is 0.612,  and for SDG 9, it is 0.607' should be '0.644; for SDG 5, it is 0.648, and for SDG 9, it is 0.611'.

8. On p. 11, l. 459, the correction 'Outer'.

9. On p. 15, l. 570, the correction 'Table 7'.

10. The list of references in alphabetic order makes the search easier.

Author Response

Thank you so much for reviewing the manuscript. Please see the attachment.

Round 3

Reviewer 3 Report

The revision was successful.